# A Validation Study for the German Versions of the Feeling Scale and the Felt Arousal Scale for a Progressive Muscle Relaxation Exercise

**DOI:** 10.3390/bs13070523

**Published:** 2023-06-22

**Authors:** Kristin Thorenz, Andre Berwinkel, Matthias Weigelt

**Affiliations:** 1Department of Sport & Health, University of Paderborn, 33098 Paderborn, Germany; matthias.weigelt@uni-paderborn.de; 2Campus Bielefeld-Bethel, University Clinic for Psychiatry and Psychotherapy (EvKB), University of Bielefeld, 33617 Bielefeld, Germany; andre.berwinkel@evkb.de

**Keywords:** feeling scale, felt arousal scale, progressive muscle relaxation, affective responses

## Abstract

The aim of the present study is to prove the construct validity of the German versions of the Feeling Scale (FS) and the Felt Arousal Scale (FAS) for a progressive muscle relaxation (PMR) exercise. A total of 228 sport science students conducted the PMR exercise for 45 min and completed the FS, the FAS, and the Self-Assessment Manikin (SAM) in a pre-test–post-test design. A significant decrease in arousal (*t*(227) = 8.296, *p* < 0.001) and a significant increase in pleasure (*t*(227) = 4.748, *p* < 0.001) were observed. For convergent validity, the correlations between the FS and the subscale SAM-P for the valence dimension (*r* = 0.67, *p* < 0.001) and between the FAS and the subscale SAM-A for the arousal dimension (*r* = 0.31, *p* < 0.001) were significant. For discriminant validity, the correlations between different constructs (FS and SAM-A, FAS and SAM-P) were not significant, whereas the discriminant analysis between the FS and the FAS revealed a negative significant correlation (*r* = −0.15, *p* < 0.001). Together, the pattern of results confirms the use of the German versions of the FS and the FAS to measure the affective response for a PMR exercise.

## 1. Introduction

Progressive muscle relaxation (PMR) is based on a relaxation technique developed by Edmund Jacobson [1] that aims at general relaxation and involves the release of residual tension. Accordingly, exercisers learn to relax different muscle groups of their body, while focusing on smaller or larger body parts (e.g., on the biceps or the whole right arm). Thus, the main goal of PMR is to decrease tension and to increase relaxation. Initially, the relaxation response is perceived as predominantly physiological, which will be experienced as a relaxation response at the psychological level after a longer period of practice [2]. PMR is scientifically well-studied for physiological disease, e.g., hypertension, headache, or chronic pain (for an overview see [3]). In the research field of psychological well-being or mental health, priority has been given to studying the effect of PMR on the states of stress or anxiety and the disorder depression [4,5]. The states of stress and anxiety involve increased tension or an over-activity of the muscles (cf., [6]). Studies on the effects of PMR show a reduction in stress and a positive impact on anxiety and depression (cf., [4,5]). Overall, the focus of research on the effectiveness of PMR has been more on clinical populations and less on healthy individuals. In addition, the current measurements capture rather long-term effects of the relaxation technique [7] and included questionnaires based on a categorical approach, e.g., the Profile of Mood States (POMS; [8]), the State-Trait Anxiety Inventory (STAI; [9]), the Smith Relaxation States Inventory (SRSI; [10]), or the Relaxation State Questionnaire (RSQ; [7]). In addition to categorical models, dimensional approaches, such as the circumplex model [11,12], should also be used to examine the effects of relaxation techniques. While measurements based on categorical approaches capture mood (e.g., POMS) or emotional states, such as anxiety (e.g., STAI) or relaxation (e.g., SRSI, RSQ), measurements based on dimensional approaches classify affective responses. A major difference between categorical and dimensional approaches is the scope of the questionnaires. Measurements with a categorical approach record multiple adjectives for a single category, whereas an assessment with a dimensional approach requires only one pair of adjectives (e.g., pleasant–unpleasant) to cover a certain dimension. Expressed differently, while the categorical approach requires several items, the dimensional approach uses only one item. 

The circumplex model orthogonally combines the valence dimension and the activation dimension (arousal), resulting in four quadrants: unactivated–pleasant affect, unactivated–unpleasant affect, activated–unpleasant affect, and activated–pleasant affect [12]. Within this model, the affective responses to physical exercise have been measured with two single-item scales, which are very economical and easy to use. These are the Feeling Scale (FS; [13]) for the valence dimension (from unpleasant to pleasant feelings) and the Felt Arousal Scale (FAS; [14]) for the arousal dimension (from low activation to high activation). For example, Ekkekakis et al. [15] used these scales to examine the affective responses for a high-intensity exercise (i.e., running on a treadmill until subjective exhaustion) and found that an increase of arousal during the stepwise increase of physical load, as assessed with the FAS, led to feelings of displeasure at the end of the treadmill ergometer task, as shown with the FS. Within the circumplex model, this emotional state reflects an activated–unpleasant affect [12]. For an exercise of moderate intensity (i.e., a bicycle ergometer task at 60% VO_2max_), van Landuyt et al. [16] reported an increase of feelings of pleasure when participants rode the bicycle for 30 min at a speed at which they felt comfortable. This emotional state reflects an activated–pleasant affect [12]. With regards to PMR, the (desired) decrease in activation should result in moving towards an emotional state of an unactivated–pleasant affect [12], which can be characterized by the experience of relaxation and calmness [17]. 

Such a pleasant affective state after performing a relaxation exercise can also be expected according to the dual-mode model (DMM; [18]). As a conceptual framework, the DMM makes predictions about the direction of affective valence (pleasant or unpleasant), which is based on the dominant processing of either cognitive or interoceptive cues, depending on the intensity of the exercise. In this regard, Brand and Kanning [19] and Bok et al. [20] summarized that an exercise intensity above the ventilatory threshold (VT) with the need of a high supply of energy provided predominantly by the anaerobic metabolism is perceived as a rather unpleasant affective state due to the dominant processing of interoceptive cues (e.g., high breathing frequency, strong muscle tension). At an exercise intensity around the VT, the aerobic metabolism is supplemented by the anaerobic metabolism to provide energy. Within this range of energy supply, the cognitive appraisal of the physical activity depends on whether the physical exertion is experienced as manageable (appraisal more pleasant) or exceeds personal capacity (appraisal more unpleasant). The supply of energy for intensities below the VT is based on aerobic metabolic processes and are generally (low interindividual variability) perceived as positive (pleasant). While Brand and Kanning [19] and Bok et al. [20] agree about the different forms of energy supply at different levels of exercise intensity, they disagree about the assignment of the processes responsible for the resulting affective responses below the VT. Brand and Kanning [19] attribute it to the processing of interoceptive cues and Bok et al. [20] to the processing of cognitive cues. 

### Purpose of the Present Study

The present study aims to extend the existing validations of the FS and the FAS for a high-intensity bicycle ergometer task [21] and for a moderate-intensity jogging exercise [22] to a relaxation exercise (i.e., progressive muscle relaxation, PMR). This follows the recommendation for an ongoing validation of measurement instruments when these are used in a new way or new context, because, in these cases, “*evidence is needed to show that the scale scores are valid representation of the construct*” [23] (p. 375). To this end, participants performed in a PMR exercise for 45 min, during which they were instructed to tense and relax the muscles of different body parts via an audio CD according to the PMR procedure by Hainbuch [24]. Before and after the PMR exercise, they completed the FS and the FAS by Maibach et al. [21], as well as the Self-Assessment Manikin (SAM) by Bradley and Lang [25] for a self–other comparison to prove the construct validity of the two single-item scales, the FS and the FAS, respectively.

The following hypotheses were derived: To satisfy the criteria of construct validity for the German versions of the FS and the FAS, respectively, significant positive correlations are expected between any two (sub)scales measuring the same construct (supporting convergent validity; Hypothesis 1), whereas no significant correlations or negative correlations should be found between any two (sub)scales of different constructs (indicating discriminant validity; Hypothesis 2). To further compare the affective responses of the PMR exercise with the results of Maibach et al. [21], the descriptive statistics of the mean values after the exercise (i.e., post-test values) will be examined for the FS and the FAS. In the present study, participants should experience the PMR exercise as more pleasant and less strenuous than the participants who performed the bicycle ergometer task in Maibach et al. [21] due to the different intensity levels and the associated different metabolic processes of energy supply in terms of the dual-mode model (cf., [18,19,20]). That is, on average, the mean post-test values of the FS were expected to be higher (Hypothesis 3) and the mean post-test values of the FAS were expected to be lower (Hypothesis 4) for the PMR exercise than for the high-intensity exercise by Maibach et al. [21]. The magnitude of change of the affective responses is hypothesized to be smaller for the valence dimension (Hypothesis 5) and for the arousal dimension (Hypothesis 6). During the PMR exercise, the focus is on relaxation awareness and tension release, but at the end, all muscle groups that were relaxed are contracted again [24]. In contrast, participants in the study by Maibach et al. [21] experienced a continuous stepwise increase of physical load during the bicycle ergometer task. In the present study, the affective responses are captured according to a whole-body activation formula, which would be comparable to a cool-down phase after a high-intensity bicycle ergometer task, and knowing that a positive change for the valence dimension can be expected only after repeated practice of the relaxation technique, more participants will perceive no change in their affective responses after the PMR exercise, which should be reflected in a higher number of zero variations for the FS and the FAS, as compared to bicycle ergometer task in Maibach et al. [21]. However, for those participants who display a change in their affective responses, the direction of change from pre-test to post-test should be positive for the FS but negative for the FAS as a direct consequence of the relaxation experience (Hypothesis 7).

## 2. Materials and Methods

### 2.1. Participants

A total of 240 students participated in the PMR exercises. Twelve participants had to be excluded due to missing values (see Figure 1 for a flow chart of the data collection and analysis procedures). Therefore, 228 students (137 females; age = 21.9 ± 2.3 years) were analyzed for the study. All participants were sport science students at the University of Paderborn. The study was carried out in several bachelor courses on sport psychological training (from 2017 until 2019). Importantly, receiving course credits did not depend on students’ participation in the study, which was therefore voluntary. In this regard, the data could not be traced back to the individual participant, as the registration and the data collection for the study were anonymized using a self-generated code. The study was approved by the university’s local ethics committee.

### 2.2. Measurement

Three questionnaires were used to assess the affective responses along the dimensions of valence and arousal. These questionnaires included the German translations of the Feeling Scale, the Felt Arousal Scale [21] and the Self-Assessment Manikin [25]. All questionnaires were presented as paper-and-pencil test versions before and after the exercise (i.e., exploiting a pre-test–post-test design).

Feeling Scale (FS; German version [21]): The FS is a numerical bipolar 11-point rating scale measuring the current mood on the valence dimension. The odd numbers and zero are verbalized. The scale ranges from −5 (“very bad“), −3 (“bad“), −1 (“fairly bad“), 0 (“neutral“), +1 (“fairly good“), +3 (“good“), to +5 (“very good”). The original scale was developed in English by Hardy and Rejeski [13].

Felt Arousal Scale (FAS; German version [21]): The FAS is a numerical 6-point rating scale measuring affective responses on the arousal dimension from 1 (“low arousal”) to 6 (“high arousal”). The original scale was developed in English by Svebak and Murgatroyd [14].

Self-Assessment Manikin (SAM; [25]): The SAM is a non-verbal pictorial assessment technique in which five manikins are presented to measure the affective responses on a scale from 1 to 5 in each of the three dimensions of emotional state: pleasure (subscale SAM-P), arousal (subscale SAM-A), and dominance (subscale SAM-D). In the pleasure dimension (subscale SAM-P), the manikins range from happy smiling (5) to unhappy frowning (1). In the arousal dimension (subscale SAM-A), the manikins range from wide-eyed excitement (5) to sleepy relaxed (1). Please note that in the present study, the dimension dominance was assessed for completeness but was not analyzed, since it is not feasible to assess the construct validity of the FS and the FAS.

### 2.3. Design

The study was conducted on two days. Participants were introduced to the study protocol, the issue of voluntary participation, and the conduct regarding anonymous data storage during the first meeting. All participants then signed the informed consent form at the beginning of the second meeting. Thereafter, they underwent the progressive muscle relaxation (PMR) exercise for 45 min in the second meeting. The PMR exercise took place in an indoor gymnasium. Before beginning the PMR exercise, participants completed the three paper-and-pencil questionnaires (i.e., pre-test), starting with the SAM, followed by the FAS and the FS. Then, they laid down in a supine position on exercise mats and followed the PMR instructions by Hainbuch [24], which were presented via one loudspeaker and an audio CD. The PMR exercise included twelve different exercises, covering the following body areas: face, neck, shoulders, and torso, as well as upper and lower extremities. After the PRM exercise, participants completed the questionnaires again (i.e., post-test) in the same order. The whole session lasted for about 55–60 min.

### 2.4. Data Analysis

Of the 240 participants who completed the PMR exercise, 12 participants had to be excluded because of missing values. Therefore, the final sample of the current study included 228 participants. The data analysis was carried out with the data software IBM SPSS Statistics for Windows, version 28.0 (Armonk, NY, USA). For construct validity, the pairs of the same construct (signifying convergent validity) and the pairs of different constructs (indicating discriminant validity) were considered. Accordingly, correlation analyses between the FS and the SAM-P for the valence dimension and between the FAS and the SAM-A for the arousal dimension were performed (for convergent validity), as well as between the pairs of different constructs (FS and SAM-A; FAS and SAM-P) for discriminant validity. In addition, the average for the post-test, the magnitude of change, and the direction of change between the pre-test and post-test, as well as the number of zero variations for all (sub)scales, were examined to directly compare the present data with the findings of Maibach et al. [21].

## 3. Results

### 3.1. Correlation of (Sub)Scales 

Table 1 displays the correlations between two (sub)scales measuring the same construct and between two (sub)scales capturing different constructs. As expected, the FS and the SAM-P (*r* = 0.67, *p* < 0.001, R^2^ = 0.45) and the FAS and the SAM-A (*r* = 0.31, *p* < 0.001, R^2^ = 0.10) were positively correlated, which signifies convergent validity. These correlations represent a large effect size for the pleasure dimension and a moderate effect size for the arousal dimension (cf. [26]). Also, the FS and the SAM-A (*r* = −0.05) and the FAS and the SAM-P (*r* = −0.09) did not correlate significantly, which supports discriminant validity. In addition, there was a significant negative correlation of a small effect size between the FS and the FAS (*r* = −0.15, *p* < 0.001, R^2^ = 0.02). When comparing the effect size of these correlations with the ones reported by Maibach et al. [21], the correlation coefficients for the valence dimension (*z* = 0.90, *p* > 0.05) and for the arousal dimension (*z* = 1.75, *p* > 0.05) were of a similar size (that is, no significant differences between the two studies), but the correlation coefficients were significantly smaller for the discriminant analysis between the FS and the FAS (*z* = 5.18, *p* < 0.001).

### 3.2. Analysis of Affective Responses

Table 2 shows the mean pre-test and post-test values (with standard deviations) and the ranges of the answers for the different (sub)scales. For the valence dimension, the means of the affective response after the PMR intervention (i.e., post-test) were for FS_post_ = 2.39 ± 1.58 and for SAM-P_post_ = 3.86 ± 0.72. The magnitude of change from pre-test to post-test was significant for the FS_change_ = 0.45 [*t*(227) = 4.748, *p* < 0.001, *d* = 0.31] and for the SAM-P_change_ = 0.11 [*t*(227) = 1.963, *p* < 0.005, *d* = 0.13]. The number of participants showing zero variations was 74 for the FS and 119 for the SAM-P (see Table 3). For the arousal dimension, the means of the affective response after the PMR intervention (i.e., post-test) were for FAS_post_ = 2.16 ± 1.00 and for SAM-A_post_ = 2.04 ± 0.85. The magnitude of change from pre-test to post-test was significant for the FAS_change_ = −0.59 [*t*(227) = 8.296, *p* < 0.001, *d* = 0.55] and for the SAM-A_change_ = −0.46 [*t*(227) = 8.116, *p* < 0.001, *d* = 0.54]. The number of participants showing zero variations was 77 for the FAS and 96 for the SAM-A (see Table 3).

## 4. Discussion

The goal of the present study was to confirm the construct validity of the German versions of the Feeling Scale (FS) and the Felt Arousal Scale (FAS) by Maibach et al. [21] for a progressive muscle relaxation (PMR) exercise. To this end, participants performed the PMR exercise for 45 min and completed a battery of questionnaires (including the FS, the FAS, and the SAM) before and after the PMR exercise. The results of the correlation analyses for construct validity (i.e., comparing any two (sub)scales of the same and of different dimensions) and the analyses of the affective responses (i.e., the post-test average, the magnitude of change, and the direction of change between pre-test and post-test, as well as the number of zero variations for all (sub)scales) confirm the use of the German versions of the FS and the FAS by Maibach et al. [21] to measure the affective responses for a PMR exercise.

### 4.1. Analysis of Correlation 

The interpretation of the results for the correlation to prove the construct validity follows the criteria of the Multitrait–Multimethod (MT-MM) analysis [27]. A high effect size (interpreted according to Cohen [26]) was demonstrated for the significant positive correlation between the FS and the SAM-P and a moderate effect size for the significant positive correlation between the FAS and the SAM-A. Accordingly, convergent validity is satisfied, and Hypothesis 1 can be confirmed. When comparing the effect size of these correlations with the ones reported by Maibach et al. [21], the correlation coefficients for the valence dimension (*z* = 0.90, *p* > 0.05) and for the arousal dimension (*z* = 1.75, *p* > 0.05) were of similar size (that is, no significant differences between the two studies). For the pairs of different constructs, only the negative correlation between the FS and the FAS was significant. This negative correlation between the level of arousal and the pleasure experience reflects the relaxation response, i.e., the decrease in arousal resulting in an increase of pleasure [28,29]. Within the circumplex model, this pattern of results can be mapped to the unactivated–pleasant affective state ([12]; Figure 2), which can be characterized by the experience of relaxation and calmness [17].

Regarding discriminant validity, all three criteria of the MT-MM analysis [27] are fulfilled. First, the correlation between the different pairs of constructs using the same method (here, FS and FAS) is lower than the correlation for the convergent validity. Second, the correlation between the different pairs of constructs of different methods (here, FS and SAM-A, FAS and SAM-P) are lower than the correlations for the same pairs of the convergent validity. Third, the correlations follow the same pattern, e.g., the same direction of the correlation coefficient (plus or minus) for the same pairs of constructs. Therefore, Hypothesis 2 is confirmed and the construct validity of the German translations of the FS and the FAS [21] can be assumed for the PMR exercise based on the correlation analyses performed in the present study.

### 4.2. Variability of Answers

The post-test value for the FS is higher for the PMR exercise (FS_post_ = 2.39) than for the high-intensity bicycle exercise (FS_post_ = −0.29) by Maibach et al. [21] and thus Hypothesis 3 can be confirmed. A positive post-test value in the pleasant range was expected because the intensity level of the PMR exercise should be below the VT for all participants, and physical activity below the VT is considered as pleasant, which can also be supported by the DMM [18,19,20]. Unfortunately, this result cannot be compared to other studies examining the affective responses for PMR exercises because previous studies used categorical questionnaires, such as the Profile of Mood States—Adolescents (e.g., [30]), the Smith Relaxation States Inventory (e.g., [31]), or the State–Trait Anxiety Inventory (e.g., [32]), or used the FS but did not report the values for the PMR exercise [33], or investigated other exercises, e.g., below or around the VT [16,34]. Hypothesis 4 can also be confirmed because the post-test value for the FAS is lower for the PMR exercise (FAS_post_ = 2.16) than for the bicycle ergometer task (FAS_post_ = 4.73) by Maibach et al. [21]. A reduction in arousal was expected for the relaxation technique [1,35]. Like for the results reflecting the pleasure dimension, the FAS post mean values cannot be compared to previous studies because these looked primarily for neurophysiological effects in a clinical setting and did not measure the subjective perceptions of arousal (see review article, e.g., [3]). 

In the present study, the magnitude of change between pre-test and post-test was 3.65 points smaller for the FS and was 1.68 points smaller for the FAS than in Maibach et al. [21]. Thus, Hypothesis 5 and Hypothesis 6 can be confirmed. A stronger influence on the valence dimension by a PMR exercise only appears after several exercise sessions [2]. During the PMR exercise, a general relaxation is provoked, even until the release of residual tension [1], while participants are activated at the end of PMR. That is, because at the end, participants are always instructed to tense all muscle groups again and are brought back to full consciousness [24]. As expected, the direction of change was positive for the FS and negative for the FAS (Hypothesis 7); this is because performing a PMR exercise causes a decrease in arousal, which has a positive effect on the valence dimension [28,29]. This pattern of results is also reflected in the negative correlation between the valence and arousal dimension discussed above.

The percentage of the zero variations shows similar proportions for the FS (32.5%) and the FAS (33.8%). In the study of Maibach et al. [21], only 9.8% of the participants (*n* = 8) displayed no variation from pre-test to post-test for the FS, whereas in the present study, 32.5% of the participants (*n* = 74) responded without variations for the FS. It is expected that, after regular practice of PMR, the proportion of zero variations will also reduce for the valence dimension [2]. For the arousal dimension, only two participants (2.4%) in the Maibach et al. [21] study showed no change for the FAS, although the load was steadily increasing. In the present study, participants experienced a constant alternation between tension and relaxation and whole-body activation at the end of the PMR exercise, which explains the higher value of zero variations of 33.8% of the participants (*n* = 77) for the FAS.

A limitation of the present study is that only two observations were made for each participant, one before (i.e., pre-test) and one after (i.e., post-test) the PMR intervention, respectively. However, it did not seem appropriate to interrupt the relaxation exercise in between (i.e., peri-test) because this would have certainly influenced the (desired) relaxation response. During the PMR exercise, it is important to keep the focus of the participants on the different muscle groups to decrease tension and to increase relaxation. Reflecting on their own emotional state in between would have drawn the focus away from the relaxation exercise, interrupting the relaxation response. This was not intended and we therefore opted for a pre-test–post-test design.

## 5. Conclusions

Together, the pattern of results of the present study confirms the construct validity for the German versions of the FS and the FAS [21] for a PMR exercise for a sample of sport science students. Thus, the single-item questionnaires FS and FAS are efficient diagnostic tools to capture the affective responses following exercises in terms of the two-dimensional approach [34]. Both scales are especially attractive (1) because of the fast and economical way in which they can be used and (2) because of the easy and straightforward way in which the affective responses can be interpreted and translated into affective states within the circumplex model [16,18,36]. Future studies could examine how long the relaxation response lasts by adding more post-tests after the intervention (e.g., after 5 min, 10 min, 30 min, 1 h). Also, testing the use of both scales in a clinical setting or for different populations (e.g., children, elderly people) could be of interest for future research.

## Figures and Tables

**Figure 1 behavsci-13-00523-f001:**
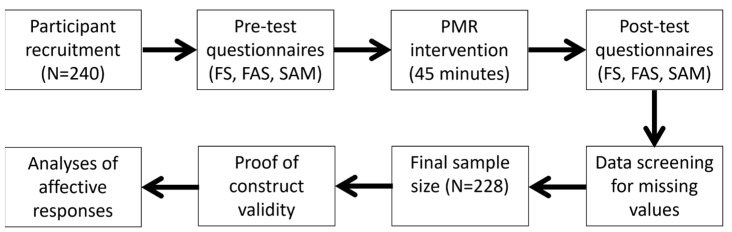
Flow chart for data collection and analysis procedures. FS = Feeling Scale, FAS = Felt Arousal Scale, and SAM = Self-Assessment Manikin.

**Figure 2 behavsci-13-00523-f002:**
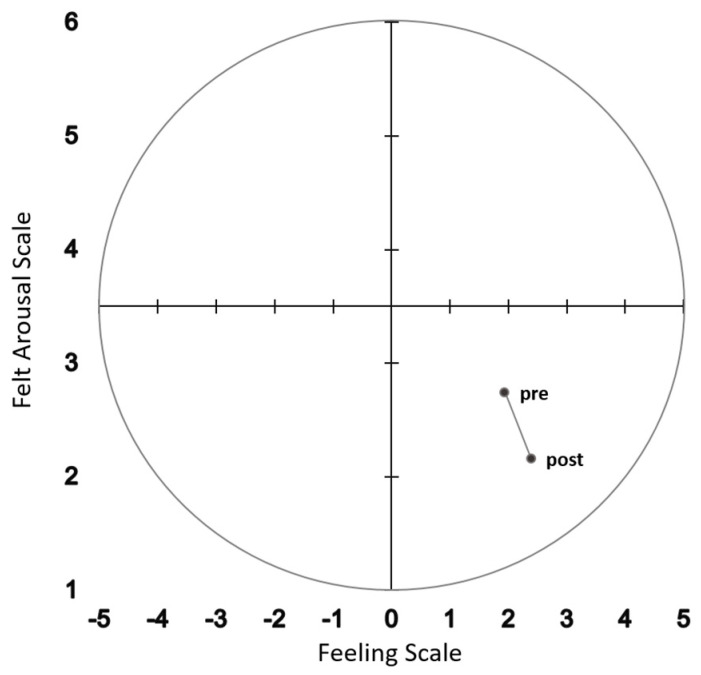
Affective response before (pre) and after (post) the progressive muscle relaxation (PMR) exercise in accordance with the two-dimensional model by Ekkekakis and Petruzzello [12].

**Table 1 behavsci-13-00523-t001:** Correlations between the different (sub)scales for the magnitude of change between pre-test and post-test (N = 228).

Variables	FS	FAS	SAM-P	SAM-A
FS	-			
FAS	−0.15 **	-		
SAM-P	0.67 **	−0.09	-	
SAM-A	−0.05	0.31 **	0.02	-

Note: FS = Feeling Scale, FAS = Felt Arousal Scale, SAM-P = pleasure dimension of Self-Assessment Manikin, SAM-A = arousal dimension of Self-Assessment Manikin. ** The correlation is significant at the level of 0.01 (two-sided).

**Table 2 behavsci-13-00523-t002:** Descriptive statistics of the pre-test and post-test data (N = 228).

Scales Used	Range of the Scale	Mean Values	Standard Deviations	Range of Answers
FS [pre]	−5–5	1.94	1.53	−4–5
FS [post]	−5–5	2.39	1.58	−3–5
FAS [pre]	1–6	2.74	0.99	1–5
FAS [post]	1–6	2.16	1.00	1–5
SAM-P [pre]	1–5	3.75	0.71	1–5
SAM-P [post]	1–5	3.86	0.72	1–5
SAM-A [pre]	1–5	2.50	0.81	1–4
SAM-A [post]	1–5	2.04	0.85	1–4

Note: FS = Feeling Scale, FAS = Felt Arousal Scale, SAM-P = pleasure dimension of the Self-Assessment Manikin, SAM-A = arousal dimension of the Self-Assessment Manikin.

**Table 3 behavsci-13-00523-t003:** Descriptive statistics of the zero variations and the direction of change from pre-test to post-test (N = 228).

Variables	Zero Variation *n* (in %)	Increase *n* (in %)	Decrease *n* (in %)
FS	74 (32.5)	114 (50.0)	40 (17.5)
FAS	77 (33.8)	24 (10.6)	127 (55.6)
SAM-P	119 (52.2)	68 (29.8)	41 (18.0)
SAM-A	96 (42.1)	25 (11.0)	107 (46.9)

Note: FS = Feeling Scale, FAS = Felt Arousal Scale, SAM-P = pleasure dimension of Self-Assessment Manikin, SAM-A = arousal dimension of Self-Assessment Manikin.

## Data Availability

The data presented in this study will be made available upon reasonable request from the corresponding author.

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
