# Peer review of "A Validation Study for the German Versions of the Feeling Scale and the Felt Arousal Scale for a Progressive Muscle Relaxation Exercise"

_behavsci, 2023, doi:10.3390/bs13070523_

Round 1

Reviewer 1 Report

Dear Authors, 

Thanks for your quality work regarding this manuscript. In general, the reviewer appreciates the contributions of this study to the fields of sports and exercise. However, there are some suggestions that may be taken into authors' consideration.

1. The introduction presents a comprehensive background of the study. It might be better to read if a conceptual framework can be performed in this study.

2. Due to several approaches being applied. A Flow chart is recommended for the data collection and analysis procedure. 

For the comments on the English language quality, there are some minor suggestions from the reviewer.

1. Formal English writing should be expected in a scientific research paper. The authors should carefully examine and revise the sentences throughout the manuscript, such as in lines 130 to 131 on page 3: "All participants were sports science students at the University of XXX."

2. Some words or similar information were repeatedly presented, it's a bit wordy.

Reviewer 2 Report

First of all, I would like to thank the journal for allowing me to review this article. Although the article has a good basis and is well argued, there are some aspects that should be modified before publication in the journal.

Introduction

The theoretical justification for the introduction is well argued, however, it is a bit terse. I recommend the authors to expand the information, mainly on the scales that have been previously used to assess the aim variables of the research.
In addition, one aspect that concerns me a lot is that I find the introduction rather disorganized. It begins by talking about the aims, but then there is a specific section on aims. These are similar, so there is repetition of information. I suggest leaving them in one place.
In addition, the aims are followed by the hypotheses. I am concerned that there are 7 hypotheses, when there is only one aim. This is excessive. I recommend reducing the number of hypotheses and increasing the number of aims so that they match (for example, 3 aims and 3 hypotheses related to them). 

Methods

How was the sample size calculated? It should be indicated. There are several methods for this (e.g., using standard deviations from previous research).

It would be appropriate to include the maximum and minimum score for each questionnaire. Also, if it allows classification in any dimension. Finally, some measure of validity and reliability (intraclass correlation coefficients, for example) could be included. 

Were the questionnaires completed in any particular order? Or was it randomized for each participant? 

What statistical program was used for the analysis? 

Results

The results are well expressed and fully understood. however, because the research aims were narrowly stated, it is difficult to follow the thread of the results. Therefore, I believe that it is necessary to change the research from the aims, as this will allow us to better follow the development of the research. 

Discussion

It is well structured and responds to all the hypotheses raised at the beginning, which is very appropriate and facilitates the reader's understanding. However, if they make the suggested modifications regarding the number of aims and hypotheses, the discussion should be modified. 

With respect to the previous research used, the discussion is well supported by and supported by them. 

Conclusions

The conclusions correctly summarize the scope of the research. However, I would include a section on limitations and possible practical applications of the research. 

Thank you very much again for allowing me to do this review.

Reviewer 3 Report

Thank you for the opportunity to review the manuscript titled “A Validation Study for the German Version of the Feeling Scale and the Felt Arousal Scale for a Progressive Muscle Relaxation Exercise”

Overall, the work is organized and comprehensively described, but there are several areas that need improvement:

1-    Please rewrite the abstract to highlight the main features, results, and conclusions of the study. Remove citations.

2-    The introduction is a little awkward. It starts with the aim and then restate the aim several times.    Contextual information is much needed there. Why the examined scales need to be validated for the German population? What are the cross-cultural differences that can be claimed to make a direct use for the translated versions invalid?  What is the significance of this work? Why FS and FAS not other available instruments?

4-    There needs to be some theoretical framework for this study. At least something to guide the study design and proposed hypotheses.

5-    The title, abstract, methods and conclusions must clearly state that this study included a validation for FS and FAS among university sport students.

6-    Were the scales already available in German language? If translated plz clarify the translation process.

7-    Who collected the data, when, how, where?

8-    Study limitations need to be well acknowledged.

Thank you

None

Round 2

Reviewer 1 Report

The revision looks better, and congratulations to the authors! Great job!

Author Response

We thank you again for your positive review of our manuscript and your comments and suggestions that have supported and improved our work.

Reviewer 2 Report

The authors have responded to all requests made. In addition, they have explained the reasons why some aspects could not be modified or included. The quality of the article has improved substantially.

Author Response

Thank you again for your comments, suggestions, and follow-up questions about our study, which were helpful in improving our work.

Reviewer 3 Report

I thank the authors for their efforts addressing reviewers comments. However, I still did not see a response with regards to the rational of the study. In particular, (1) What is the significance of this work? If this study was not conducted, what gaps in the literature remain? Don't researchers already use the translated versions?

(2) I still did not see the added value of this study when compared to that by Maibach et al. (2020). Based on what did authors assume that the tools cannot be claimed valid for low intensity exercise? Currently, this entire study seem to be based only on a recommendation made by single study "Moosbrugger and Kelava [24]".

None
